# SARS-CoV-2 vaccination, booster, and infection in pregnant population enhances passive immunity in neonates

Elisabeth A. Murphy[1], Camila Guzman-Cardozo[2], Ashley C. Sukhu[3], Debby J. Parks ®[2], Malavika Prabhu[4], Iman Mohammed[4], Magdalena Jurkiewicz[1], Thomas J. Ketas[5], Sunidhi Singh[6], Marie Canis[2], Eva Bednarski[2], Alexis Hollingsworth[6], Embree M. Thompson[6], Dorothy Eng[3], Paul D. Bieniasz ®[2,7], Laura E. Riley[4], Theodora Hatziioannou ®[2,8] ✉ & Yawei J. Yang ®[1,3,8] ✉

The effects of heterogeneous infection, vaccination and boosting histories prior to and during pregnancy have not been extensively studied and are likely important for protection of neonates. We measure levels of spike binding antibodies in 4600 patients and their neonates with different vaccination statuses, with and without history of SARS-CoV-2 infection. We investigate neutralizing antibody activity against different SARS-CoV-2 variant pseudotypes in a subset of 259 patients and determined correlation between IgG levels and variant neutralizing activity. We further study the ability of maternal antibody and neutralizing measurements to predict neutralizing antibody activity in the umbilical cord blood of neonates. In this work, we show SARS-CoV-2 vaccination and boosting, especially in the setting of previous infection, leads to significant increases in antibody levels and neutralizing activity even against the recent omicron BA.1 and BA.5 variants in both pregnant patients and their neonates.

In March 2020 New York City became the epicenter of the COVID19 pandemic and residents of New York City experienced high rates of SARS-CoV-2 exposure[1]. Repeated waves of SARS-CoV-2 infection, as well as the deployment of mRNA and adenovirus vectored vaccines that became readily available in the United States in early 2021 means that the population have experienced heterogeneous exposures to SARS-CoV-2 antigens[2–4].

While pregnancy does not increase the likelihood of becoming SARS-CoV-2 infected, infection during pregnancy increases the risk of morbidity and mortality compared to the non-pregnant population[5,6]. Conversely, studies of vaccination during pregnancy have not identified any increased risks or any adverse outcomes to the pregnancy, the placenta or fetal development[7–13]. Thus, women have been advised to receive vaccines, and multiple vaccine doses have been administered during pregnancy[14]. Vaccination during pregnancy leads to robust immunoglobulin G (IgG) antibody response in the pregnant patients, but also passive transfer of antibodies to their neonates[15–21]. In fact, vaccination during pregnancy has been shown to reduce maternal SARS-CoV-2 infections and to prevent COVID-19 hospitalization among infants born to vaccinated patients for up to six months of age[22–25]. Booster doses after completion of the primary vaccination series are recommended and have been shown to lead to a resurgence in antibody levels and the generation of neutralizing antibodies that are more effective against emergent SARS-CoV-2 variants[26–29]. However, the

[1]Department of Pathology and Laboratory Medicine, Weill Cornell Medicine, New York, NY, US. [2]Laboratory of Retrovirology, The Rockefeller University, New York, NY, US. [3]Department of Pathology and Laboratory Medicine, New York Presbyterian/Weill Cornell Medical Center, New York, NY, US. [4]Department of Obstetrics & Gynecology, Weill Cornell Medicine, New York, NY, US. [5]Department of Microbiology and Immunology, Weill Cornell Medicine, New York, NY, US. [6]Weill Cornell Medicine, New York, NY, US. [7]Howard Hughes Medical Institute, The Rockefeller University, New York, NY, US. [8]These authors jointly supervised this work: Theodora Hatziioannou, Yawei J. Yang. ✉e-mail: thatziio@mail.rockefeller.edu; yang@med.cornell.edu

benefits to the neonate born to a mother that received a full course vaccination and a subsequent booster in the setting of past infection are not clear. In particular, protection of the neonate is entirely dependent on the passive transfer of antibodies, while protection of the mother likely involves multiple components of the previous infection or vaccine elicited immune response.

Although vaccination affords high levels of protection from severe disease in adults, protection from infection is reduced and continually eroded by variants that are increasingly resistant to neutralizing antibodies[17,30,31]. The level of SARS-CoV-2 neutralizing antibody resistance has, thus far, been dramatically eroded by the emergence of the omicron variants[32–34]. The omicron BA.1, BA.2, and BA.5 variants initiate infection waves across the world with BA.2 and BA.5 derivatives currently further diversifying with the accumulation of further neutralizing antibody resistant mutations. In addition, there is continued vaccination and booster hesitancy[35]. This scenario has resulted in a diverse landscape of variant circulation amongst different populations.

Binding and neutralizing antibody levels against the spike protein and its receptor-binding domain (RBD) have been shown to be predictors of protection against symptomatic infection, and neutralizing antibody levels are associated with vaccine efficacy in adults; in turn, neutralizing antibody levels have been found to correlate well with levels of antibodies binding to the virus spike[36–38]. In this study, we investigated the impact of vaccination on antibody levels in a cohort of pregnant patients that had received different number of vaccine doses, primarily mRNA, and boosters and had experienced or not experienced SARS-CoV-2 infection. We also measured neutralizing antibodies against the ancestral and omicron SARS-CoV-2 variants and determined the degree to which potentially protective antibody responses were passively transferred to their neonates.

## Results

### Patient Population

Our study included 4600 patients of whom 2999 either had a history of vaccination or infection or both, as well as 1601 patients who did not have a history of vaccination or infection as controls (Table 1). The median (IQR) maternal age at the time of delivery was 35 (6) years and the median (IQR) gestational age at delivery was 39.3 (1.50) weeks (Table 1, Supplementary Table 1).

The study had 2109 patients who did not receive any doses of a SARS-CoV-2 vaccine, and 2491 patients who had received at least one dose of a SARS-CoV-2 vaccine (Vx) (Table 1, Supplementary Table 1). The patients that received at least one dose of a vaccine were vaccinated between December 15, 2020, and March 4, 2022. The median (IQR) gestational age at vaccine initiation was 14.4 (29.9) weeks (Table 1). Vaccination status of our cohort included 245 patients that had started but not completed their vaccination course (PartVx), 1578 patients that had completed their vaccination course at least 14 days prior (FullVx), and 668 patients that had received a booster dose after a complete vaccination course (BoostVx) (Materials and Methods for categorization details, Table 1, Fig. 1a).

Our study included 3589 patients with no history of SARS-CoV-2 infection and 1011 patients that had a documented history of SARS-CoV-2 infection (Inf) (Table 1, Fig. 1a). Of those with a positive history of infection, 516 had a specific documented date of a clinical test for SARS-CoV-2 infection, and the documented infections occurred between February 1, 2020, and April 27, 2022. The remainder of the patients with a positive history of infection only provided approximate time frames of when they were infected.

### IgG antibody response in SARS-CoV-2 vaccination and infection

Among the 3589 patients with no history of infection, each vaccine dose received resulted in a significant increase in IgG level (NoVx vs PartVx $p < 2.22e\text{-}16$; PartVx vs FullVx $p = 3.7e\text{-}9$; FullVx vs BoostVx

**Table 1 | Demographics, vaccination data, and history of Infection on complete cohort of patients**

| | |
|---|---|
| Patient samples, $n$ | 4600 |
| Age, Median (IQR) years | 35 (6) |
| Gestation age at delivery, Median (IQR) weeks | 39.3 (1.5) |
| Umbilical cord blood samples captured, $n$ | 2706 |
| **Vaccination Cohorts** | |
| No history of infection | |
| Not vaccinated (NoVx) | 1601 |
| Partially vaccinated (PartVx) | 194 |
| Fully vaccinated (FullVx) | 1311 |
| Boosted (BoostVx) | 483 |
| No history of infection | |
| Not vaccinated (NoVx/Inf) | 508 |
| Partially vaccinated (PartVx/Inf) | 51 |
| Fully vaccinated (FullVx/Inf) | 267 |
| Boosted (BoostVx/Inf) | 185 |
| **Vaccination course received, $n$** | |
| Pfizer-BioNTech | 1800 |
| Moderna | 616 |
| Johnson & Johnson | 64 |
| Mixed dosing | 13 |
| **Booster Vaccine received, $n$** | |
| Pfizer-BioNTech | 481 |
| Moderna | 186 |
| Johnson & Johnson | 1 |
| **Gestational age at first vaccination dose, $n$** | |
| Before pregnancy | 719 |
| 1st trimester | 508 |
| 2nd trimester | 704 |
| 3rd trimester | 545 |
| **Gestational age at second vaccination dose, $n$** | |
| Before pregnancy | 590 |
| 1st trimester | 387 |
| 2nd trimester | 674 |
| 3rd trimester | 689 |
| **Gestational age at third vaccination dose, $n$** | |
| Before pregnancy | 3 |
| 1st trimester | 24 |
| 2nd trimester | 317 |
| 3rd trimester | 324 |

Full course of vaccination was categorized as 1 dose for Johnson&Johnson vaccination, 2 doses for Pfizer-BioNTech vaccination, and 2 doses for Moderna vaccination. Patients were listed as not having received any vaccination (NoVx), started a vaccination course but not yet 14 days post the completion of a full course of vaccination (PartVx), at least 14 days post completion of a full course of vaccination (FullVx), or received an additional booster dose after completion of a full course of vaccination (for a total of 2 doses for Johnson&Johnson, 3 doses for Pfizer-BioNTech, 3 doses for Moderna) (BoostVx). The BoostVx patients were confirmed to not have any immuno-suppressing condition or use of an immunosuppressing medication. Mixed dosing refers to patients that received a combination of Pfizer/BioNTech and Moderna for the vaccination course.

$p < 2.22e\text{-}16$), with those that received a booster dose (BoostVx) demonstrating the highest IgG levels (Fig. 1b). Among the 1011 patients that had a history of SARS-CoV-2 infection, the first vaccine dose resulted in a significant increase in IgG levels (NoVx/Inf vs PartVx/Inf $p = 1.4e\text{-}15$). Completion of the vaccination series in these patients resulted in a minor further increase (PartVx/Inf vs FullVx/Inf, $p = 0.44$), whereas receipt of a booster dose led to an additional notable increase in IgG levels (FullVx/Inf vs BoostVx/Inf $p < 2.22e\text{-}16$) (Fig. 1c).

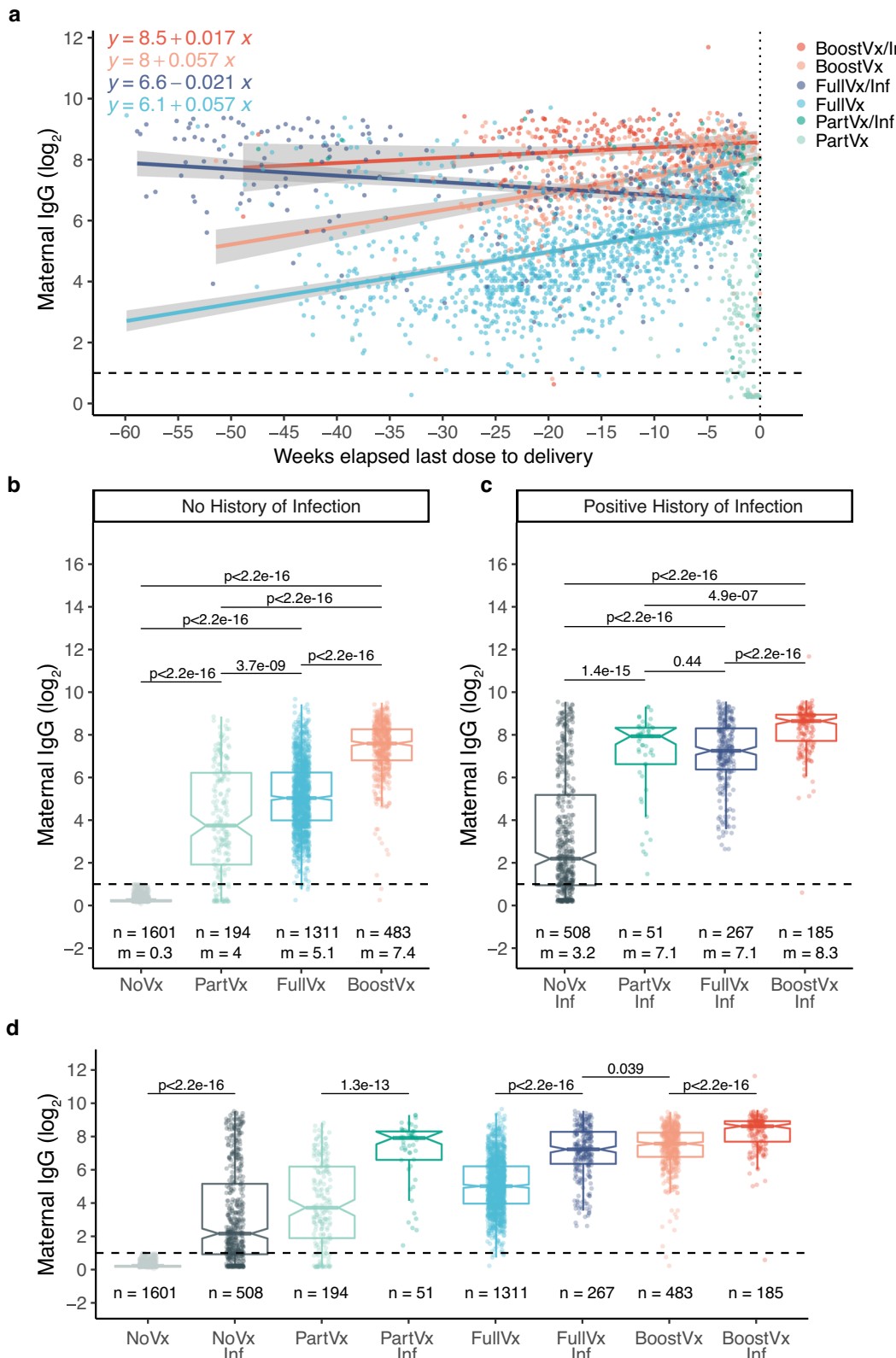

When comparing those without to those with a history of infection, the infected group had significantly higher levels of IgG in all vaccination cohorts (NoVx $p < 2.22e\text{-}16$; PartVx $p = 1.3e\text{-}13$, FullVx $p < 2.22e\text{-}16$, BoostVx $p < 2.22e\text{-}16$) (Fig. 1d). Antibody levels in the fully vaccinated and positive history of infection cohort (FullVx/Inf) were slightly lower than the fully boosted cohort without a history of infection (BoostVx) ($p = 0.039$) (Fig. 1d). Models of antibody level variation over time

showed that the BoostVx/Inf cohort had the highest antibody levels over time ($y = 8.5 + 0.017x$) and that these antibodies could be detected up to 50 weeks after the last vaccine dose was administered (Fig. 1a). Analyses of differences in IgG levels elicited between vaccination types showed that there was only a significant difference between receipt of mRNA1273 (Moderna) vaccine or the BNT162b2 (Pfizer-BioNTech) vaccine in the FullVx ($p = 1.4e\text{-}11$) and BoostVx

**Fig. 1 | Anti-S IgG response to SARS-CoV-2 vaccination relative to delivery in pregnant patients. a** All patients who received at least one dose of SARS-CoV-2 vaccination prior to delivery. Maternal anti-S IgG levels are examined relative to time elapsed since SARS-CoV-2 vaccination dose and delivery (0). The relationship between maternal anti-S IgG levels and time since last vaccination dose relative to delivery was examined by linear regression model for vaccinated and boosted patients; variables contributing to significant differences were determined by mixed ANOVA. Patients are grouped according to vaccination status at time of delivery and history of SARS-CoV-2 infection: Boosted with positive history of infection (BoostVx/Inf); Boosted with no history of infection (BoostVx); Fully vaccinated with positive history of infection (FullVx/Inf); Fully vaccinated with no history infection (FullVx); Partially vaccinated with positive history of infection (PartVx/Inf); Partially vaccinated with no history of infection (PartVx). **b**–**d** Maternal anti-S IgG levels in patients with no history of infection stratified by vaccination

status at time of delivery: Not vaccinated (NoVx); Partially vaccinated (PartVx); Fully Vaccinated (FullVx); Boosted (BoostVx). Maternal anti-S IgG levels in patients with positive history of SARS-CoV-2 infection stratified by vaccination status at time of delivery: Not vaccinated with positive history of infection (NoVx/Inf); Partially vaccinated with positive history of infection (PartVx/Inf); Fully vaccinated with positive history of infection (FullVx/Inf); Boosted with positive history of infection (BoostVx/Inf). Differences within the no history **b** and history **c** groups, as well as differences between groups with the same vaccination status **d** were examined using two-sided Wilcoxon rank-sum test. Data points are indicated as dots and plotted as boxplots (center line, median; bounds of box, the first and third quartiles; whiskers, 1.5 times interquartile range). Shaded regions indicate 95% confidence intervals for each linear relationship visualized. Source data are provided as a source data file.

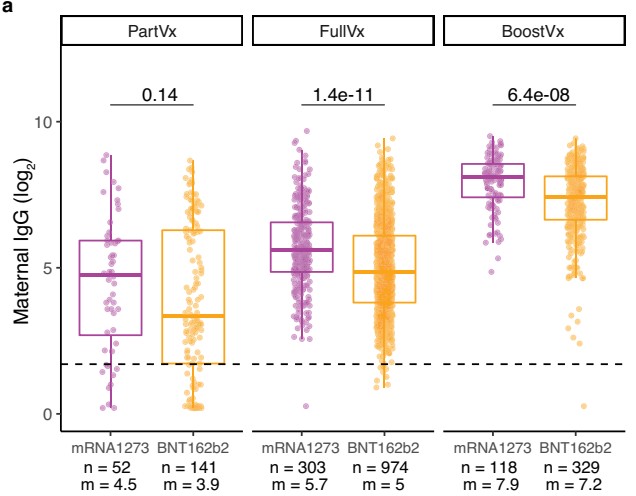

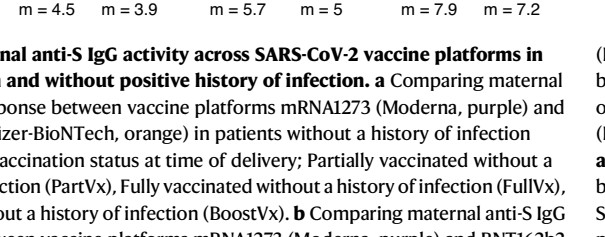

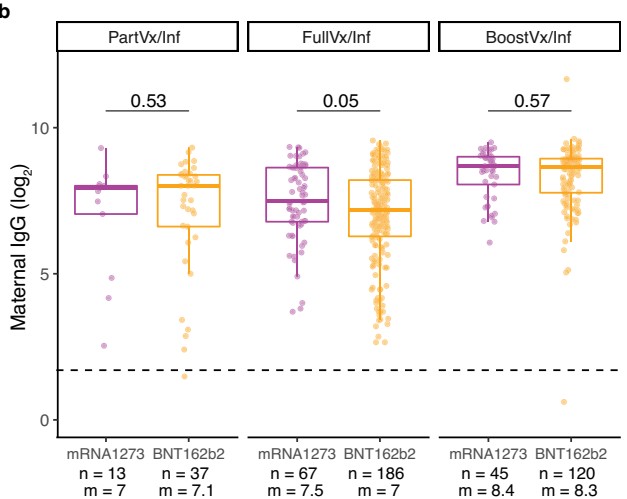

**Fig. 2 | Maternal anti-S IgG activity across SARS-CoV-2 vaccine platforms in patients with and without positive history of infection. a** Comparing maternal anti-S IgG response between vaccine platforms mRNA1273 (Moderna, purple) and BNT162b2 (Pfizer-BioNTech, orange) in patients without a history of infection stratified by vaccination status at time of delivery; Partially vaccinated without a history of infection (PartVx), Fully vaccinated without a history of infection (FullVx), Boosted without a history of infection (BoostVx). **b** Comparing maternal anti-S IgG response between vaccine platforms mRNA1273 (Moderna, purple) and BNT162b2

(Pfizer-BioNTech, orange) in patients with a positive history of infection stratified by vaccination status at time of delivery; Partially vaccinated with a positive history of infection (PartVx/Inf), Fully vaccinated with a positive history of infection (FullVx/Inf) and Boosted with a positive history of infection (BoostVx/Inf). For **a**, **b**, data points are indicated as dots and plotted as boxplots (center line, median; bounds of box, the first and third quartiles; whiskers, 1.5 times interquartile range). Significance was determined by two-sided Wilcoxon rank-sum Test. Source data are provided as a source data file.

(p = 6.4e-8) populations without a history of infection, but not in any of the infected patient populations (all $p > 0.05$) (Fig. 2a, b). Mixed ANOVA analysis demonstrated that the time elapsed between the last dose of a SARS-CoV-2 vaccine and delivery, the vaccination status, and the history of infection of the patient all significantly influence maternal IgG levels (all $p < 2.2e-16$). The precise timing of infection was only available for a small subset of patients, nevertheless we did not see a significant association between time elapsed since infection and maternal IgG levels ($p > 0.05$).

### Neutralization antibody levels in pregnant patients

Neutralization studies of a representative 259 patients with a history of infection showed that patients who received a booster dose of vaccination and also had a history of infection (BoostVx/Inf) demonstrated the highest neutralizing activity for the B.1 variant compared to all other vaccination cohorts (BoostVx/Inf compared: to FullVx/Inf $p = 0.013$; to PartVx/Inf $p = 0.013$; to NoVx/Inf $p = 8.4e-6$) (Fig. 3a, Supplementary Table 2). In contrast, for BA.1 and BA.5, the same boosted patients with a positive history of infection (BoostVx/Inf) showed significantly higher neutralizing activity only when compared to the unvaccinated cohorts (NoVx/Inf) (BA.1 $p = 2.9e-5$; BA.5

$p = 0.00093$) (Fig. 3b, c). The serum neutralizing activity against the B.1 spike variant correlated strongly with the semiquantitative IgG levels (B.1 $r = 0.8$, $p < 2.2$ e-16) (Fig. 3d). The correlation between IgG levels and neutralization of the omicron variant spikes BA.1 and BA.5 was lower than that for B.1 (BA.1 $r = 0.74$, $p < 2.2$ e-16; BA.5 $r = 0.66$, $p < 2.2e-16$) (Fig. 3e, f). This difference likely reflects the heterogeneity in the neutralizing antibody activity against omicron spikes stemming from the variability of infection history in the cohort.

All 8 patients with the highest B.1 neutralizing antibody levels (set as greater than 1 standard deviation from the mean of the cohort), were either fully vaccinated or fully boosted. In contrast, of the 35 patients with the lowest B.1 neutralizing activity 68.6% (24/35) were not vaccinated by time of delivery. 50 of the participants had high BA.1 or BA.5 neutralizing activity and 96.0% of those patients (48/50) had documented infection dates between December 1, 2021, and March 24, 2022, when the omicron BA.1 variant first appeared and quickly became the most prevalent variant in New York City (Fig. 3a & Supplementary Fig. 1a−c). Further sub-analyses of our cohort showed there were no differences detected in neutralizing activity when comparing between Moderna and Pfizer-BioNTech vaccination recipients (all $p > 0.05$) (Fig. 4a−c).

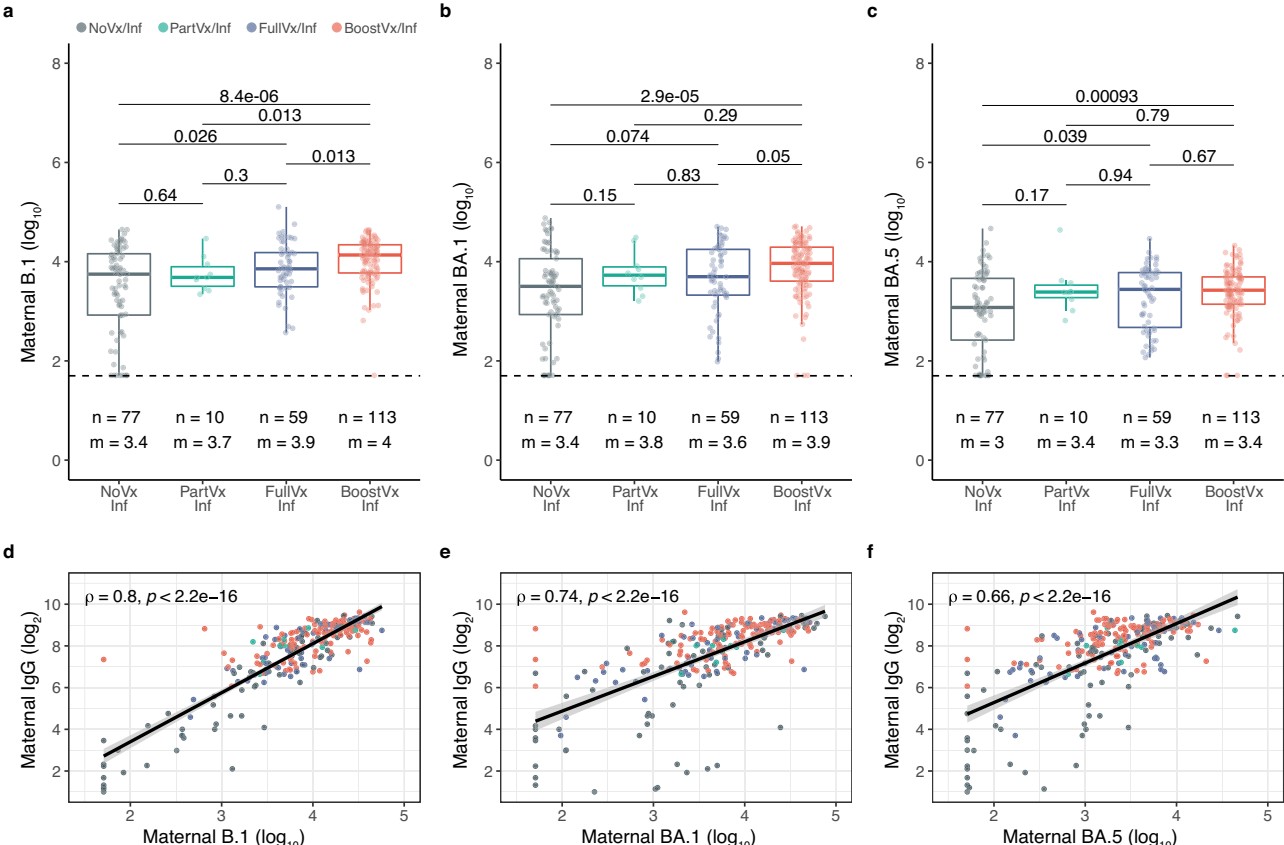

**Fig. 3 | Maternal neutralizing activity against B.1, BA.1, and BA.5 spike variants in patients with a positive history of SARS-CoV-2 infection. a–c** Neutralizing activity in patients with a positive history of SARS-CoV-2 infection stratified according to vaccination status at time of delivery: Not vaccinated with positive history of infection (NoVx/Inf); Partially vaccinated with positive history of infection (PartVx/Inf); Fully vaccinated with positive history of infection (FullVx/Inf); Boosted with positive history of infection (BoostVx/Inf). Data points are indicated as dots and plotted as boxplots (center line, median; bounds of box, the first and third quartiles; whiskers, 1.5 times interquartile range). Differences across groups were determined by two-sided Wilcoxon rank-sum test. **d–f.** Correlation between maternal anti-S IgG levels (log$_2$ transformed) and neutralizing antibody levels (log$_{10}$ transformed) in patients with positive history of SARS-CoV-2 infection. Not vaccinated with positive history of infection (NoVx/Inf); Partially vaccinated with positive history of infection (PartVx/Inf); Fully vaccinated with positive history of infection (FullVx/Inf); Boosted with positive history of infection (BoostVx/Inf). Spearman correlation analysis were carried out for each variant assayed: B.1; BA.1; BA.5. The half-maximal neutralization titers for each plasma (NT50). Shaded regions indicate 95% confidence intervals for each linear relationship visualized. Source data are provided as a source data file.

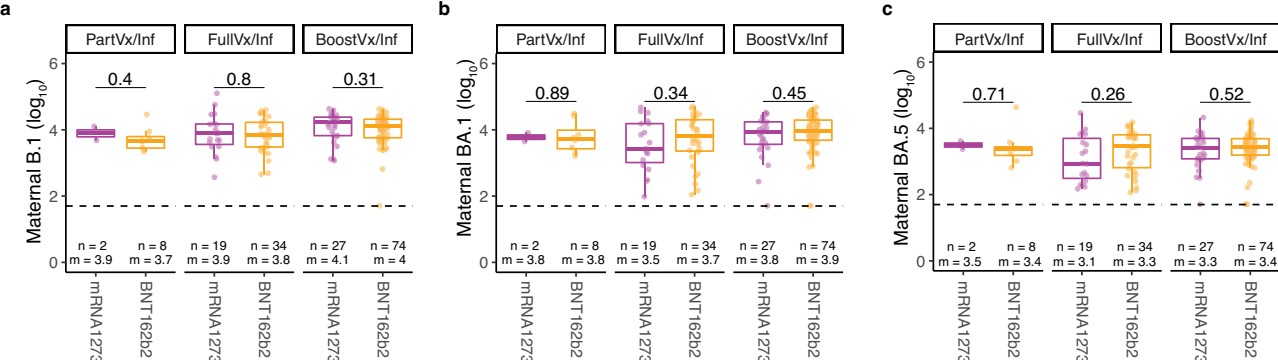

**Fig. 4 | Comparing neutralizing activity against variants across vaccine platforms in vaccinated patients. a–c** Neutralizing activity against **a** B.1, **b** BA.1, and **c** BA.5 variants in patients with a positive history of SARS-CoV-2 infection compared across vaccine platforms mRNA1273 (Moderna, purple) and BNT162b2 (Pfizer-BioNTech, orange). Patients are stratified by vaccination status at time of delivery; Partially vaccinated with a positive history of infection (PartVx/Inf), Fully vaccinated with a positive history of infection (FullVx/Inf) and Boosted with a positive history of infection (BoostVx/Inf). Data points are indicated as dots and plotted as boxplots (center line, median; bounds of box, the first and third quartiles; whiskers, 1.5 times interquartile range). Significance was determined by two-sided Wilcoxon rank-sum Test. Source data are provided as a source data file.

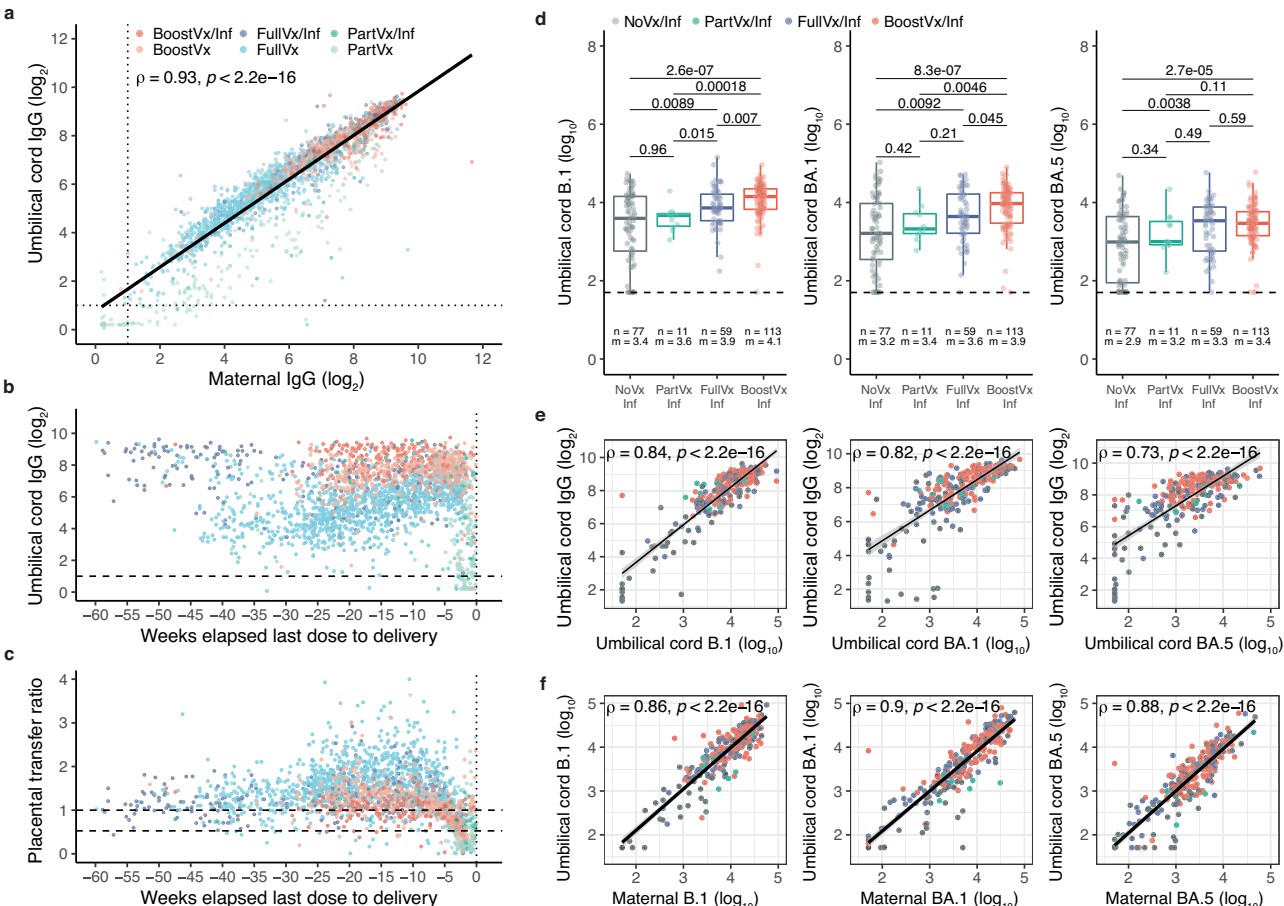

**Fig. 5 | Neutralizing activity against B.1, BA.1, and BA.5 spike variants and anti-S IgG response in neonates born to patients with a positive history of SARS-CoV-2 infection. a** Matched maternal peripheral blood over neonatal umbilical cord blood anti-S IgG levels at time of delivery. Spearman correlation analysis on log$_2$-transformed values. **b** Umbilical cord anti-S IgG levels of neonates born to patients who received at least one dose of SARS-CoV-2 vaccination prior to delivery. Umbilical cord anti-S IgG levels are examined relative to time elapsed between last SARS-CoV-2 vaccination dose and delivery. **c** Placental transfer ratios (umbilical cord anti-S IgG/maternal anti-S IgG levels) were studied relative to time elapsed between last SARS-CoV-2 vaccination dose and delivery. For **a–c** samples were grouped according to maternal vaccination status at time of delivery: Partially vaccinated with no history of infection (PartVx); Partially vaccinated with positive history of infection (PartVx/Inf); Fully Vaccinated with no history of infection (FullVx); Fully vaccinated with positive history of infection (FullVx/Inf); Boosted with no history of infection (BoostVx); Boosted with positive history of infection (BoostVx/Inf). Source data are provided as a source data file. **d** Neutralizing activity in umbilical cord blood of neonates born to patients with a positive history of SARS-

CoV-2 infection stratified according to vaccination status at time of delivery. Differences were determined by two-sided Wilcoxon rank-sum test. Data points are indicated as dots and plotted as boxplots (center line, median; bounds of box, the first and third quartiles; whiskers, 1.5 times interquartile range). **e** Umbilical cord anti-S IgG levels over neutralizing antibody levels in neonates born to patients with positive history of COVID-19 infection at time of delivery. Spearman correlation analysis were carried out for each variant assayed (B.1; BA.1; BA.5). **f** Matched maternal peripheral blood over neonatal umbilical cord blood neutralizing activity at time of delivery per variant assayed (B.1; BA.1; BA.5). Spearman correlation analysis on log$_{10}$-transformed values. For **d–f**, samples are grouped according to maternal vaccination status at time of delivery: Not vaccinated with positive history of infection (NoVx/Inf); Partially vaccinated with positive history of infection (PartVx/Inf); Fully vaccinated with positive history of infection (FullVx/Inf); Boosted with positive history of infection (BoostVx/Inf). Shaded regions indicate 95% confidence intervals for each linear relationship visualized. Source data are provided as a source data file.

## Neutralization antibody levels in neonates

Anti-S IgG levels in umbilical cord blood from 2706 neonates strongly correlated with the maternal IgG levels ($r = 0.93$, $p < 2.2e$-16) (Fig. 5a). 56 umbilical cord samples had IgG levels below the positive cutoff, and of those 56 samples, 91% (51/56) were born to partially vaccinated patients (Fig. 5b). The placental transfer ratio was above 0.5 for 93% (2248/2429) and above 1 for 73% (1785/2429) of all dyads tested (Fig. 5c).

Neutralization studies of a representative 260 number of neonates born to 259 patients with a history of infection showed that neonates born to patients that received a booster dose and had a positive history of infection (BoostVx/Inf) had the highest neutralizing activity against B.1 and BA.1 compared to other vaccination cohorts (compared to FullVx/Inf B.1 $p = 0.007$, BA.1 $p = 0.045$; to PartVx/Inf B.1

$p = 0.00018$, BA.1 $p = 0.0046$; to NoVx/Inf B.1 $p = 2.6e$-7 BA.1 $p = 8.3e$-7) (Fig. 5d). These same BoostVx/Inf neonates also had significantly higher neutralizing activity against BA.5 when compared to the NoVx/Inf cohorts ($p = 2.7e$-5) (Fig. 5d). A total of 10 of the 77 neonates born to unvaccinated patients had no detectable neutralizing activity even against B.1 (Fig. 5e). These 10 neonates were born to patients with either undetectable (8/10) or low B.1 (2/10) neutralizing activity (Fig. 5f).

The neonatal cord IgG levels correlated with neonatal cord blood neutralizing activity against the B.1, BA.1, and BA.5 spike pseudotypes (B.1 $r = 0.84$ $p < 2.2e$-16; BA.1 $r = 0.82$, $p < 2.2e$-16; BA.5 $r = 0.73$, $p < 2.2e$-16) (Fig. 5e). Importantly, there was a high correlation between the neutralizing antibody levels in neonatal cord blood and maternal peripheral blood for all spike variants assayed (B.1 $r = 0.86$, $p < 2.2e$-16;

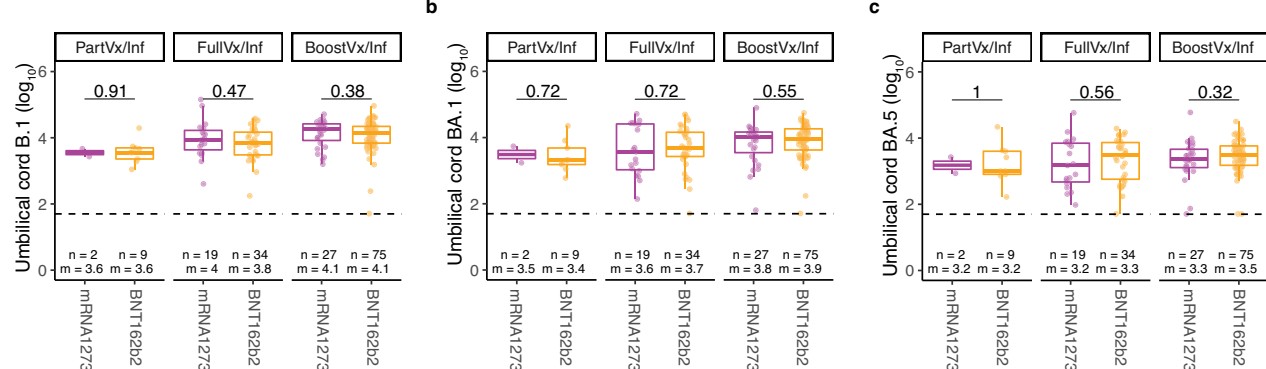

**Fig. 6 | Comparing neutralizing activity against variants across vaccine platforms in neonates of vaccinated patients. a–c** Neutralizing activity against **a** B.1, **b** BA.1, and **c** BA.5 variants in umbilical cord blood from patients with a positive history of SARS-CoV-2 infection compared across vaccine platforms; mRNA1273 (Moderna, purple) and BNT162b2 (Pfizer-BioNTech, orange). Samples are stratified by patient vaccination status at time of delivery; Partially vaccinated with a positive history of infection (PartVx/Inf), Fully vaccinated with a positive history of infection (FullVx/Inf) and Boosted with a positive history of infection (BoostVx/Inf). Data points are indicated as dots and plotted as boxplots (center line, median; bounds of box, the first and third quartiles; whiskers, 1.5 times interquartile range). Significance was determined by two-sided Wilcoxon rank-sum Test. Source data are provided as a source data file.

BA.1 r = 0.9, *p* < 2.2e-16; BA.5 *r* = 0.88, *p* < 2.2e-16) (Fig. 5f). Additional sub-analyses showed that there were also no differences in umbilical cord blood neutralizing activity between vaccination types received (Fig. 6a–c).

## Discussion

We analyzed antibody responses in a large cohort of pregnant patients with a heterogeneous history of SARS-CoV-2 vaccination and infection. Our results demonstrate that each additional immunization leads to higher total anti-S antibody production. We also showed that a history of SARS-CoV-2 infection leads to a significant increase in total antibody levels when compared to those without a history of infection and equivalent vaccination status. Accordingly, the highest levels of antibodies detected were in pregnant patients that had a history of SARS-CoV-2 infection and received all vaccine doses including a booster. These patients, with a history of infection and that received a booster vaccine dose, also had the highest neutralizing activity against B.1, BA.1, and BA.5. Patients not fully vaccinated had generally low neutralizing activity against B.1, and patients with infections during the peak omicron wave in New York City generally had high neutralizing activity against BA.1 and BA.5. We confirmed that the levels and neutralizing activity of antibodies detected in the maternal peripheral blood correlated highly with the levels and neutralizing activity of the antibodies detected in the umbilical cord blood. The majority of neonates with antibody levels below the level of detection were born to patients that did not complete a vaccination course; conversely, neonates with the highest antibody levels and neutralizing activity were born to patients that were both boosted and had a positive history of SARS-CoV-2 infection. For patients with a history of infection, we did not see a difference in neutralizing activity against B.1, BA.1, and BA.5 between vaccination type received in either the maternal peripheral blood or neonatal cord blood.

Limitations in our study include the dependence on self-reported history of infection and a fraction of the infected cohort lacked precise dates of infection or viral sequencing, and therefore we cannot deduce the variant responsible for the infection. The lack of precise timing of infection may have hindered our ability to elucidate significant association between time elapsed since infection and antibody levels as well as the potential variants of infection. In addition, the lack of timing did not allow us to sub stratify patients based on when the vaccinations and infections took place relative to one another; thus, grouping of all infected patients cannot address the direct effect of infection on antibody levels from vaccination or the direct effect of vaccination on

antibody levels from infection. While we confirmed that patients with a history of infection had detectable anti-N antibodies, we were unable to run these tests in all patients without a history of infection and it is possible that some of the patients were miscategorized in that group especially if they had mild or asymptomatic infections. Given the recommended spacing of months between the primary vaccination course and the booster dose, it is possible that the boosted cohort started vaccination earlier than the non-boosted cohort; however, our analysis has been performed with respect to the timing of the last vaccination dose received prior to birth and we have ensured that there is a good spread of patients with comparable demographics in all cohorts tested to the best of our ability. Our study did not include analysis on the gestational age at any singular vaccination receipt which is a predictive factor that could be playing a role in the level of passive transfer of antibodies; however, our patient population has a median (IQR) gestational age at birth of 39.3 (1.5) weeks and our analysis on weeks elapse from last vaccination to birth may serve as a proxy for understanding this possible confounding variable. Due to the timing of our study, unfortunately we were not able to include patients that received bivalent booster dose of vaccination.

While passive immunity provided by SARS-CoV-2 antibodies post vaccination has been reported, with correlation between maternal antibody levels and neonatal umbilical cord blood levels, prior studies did not account for both the vaccination status and the infection history of the pregnant patients nor the changing variant landscape at a large scale. Smaller studies have also reported the neutralizing activity of antibodies passed from pregnant patients to their neonates[15,16,28,30]. However, this is the first study to include such a large population of pregnant patients and their neonates, allowing us to delineate the effect of vaccination status and the effect of infection in this population. Indeed, previous studies have analyzed the antibody response in pregnant patients and neonates in the setting of vaccination or infection alone, but not at the effects of patient SARS-CoV-2 infection history and vaccination status combined while teasing out the effects of additional vaccine doses. We also correlated antibody levels with neutralizing activity not only against the B.1, but also the recent omicron BA.1 and BA.5 variants and confirmed that vaccination and particularly administration of the booster dose leads to neutralization activity against these variants. Furthermore, unlike prior studies, we show that the level of total maternal anti-S antibodies predicts neutralization activity against each of these variants in pregnant patients and neonates. The strong correlation in antibodies detected between patient and their neonates does not support the presence of any

restriction in the passage of anti-S IgG antibodies via the blood supply. Rather, it supports the notion that increasing antibody levels in patients via boosting, results in the transfer of the highest antibody levels to their neonates. The stepwise increase in B.1 neutralizing activity as patients go from unvaccinated to vaccinated to boosted, results in an equivalent increase in antibodies transferred to neonates. The levels of total anti-S IgG correlated very well with B.1 neutralizing activity, as expected since both the initial vaccinations as well as many clinical antibody detection strategies rely on reagents based on the original B.1 spike of SARS-CoV-2.

While our current study focuses on pregnant patients, our previous studies confirmed that the antibody responses in pregnant patients are comparable to non-pregnant patients[17], thus it is possible that we can extrapolate our findings beyond the pregnant patient population to offer insights into the immune response of the general population. The cumulative time elapsed since the SARS-CoV-2 epidemic started along with the emergence of SARS-CoV-2 variants with high rates of transmission have resulted in a large population of people with a history of SARS-CoV-2 infection. Vaccination and boosting in all patients, and especially in those with a history of past SARS-CoV-2 infection, lead to higher antibody levels and neutralizing activity. The beneficial impact of receiving a booster vaccine dose on the levels and neutralizing activity of antibodies detected in the umbilical cord blood also supports the recommendations of vaccination and boosting to benefit neonates that are uniquely dependent on passively transferred antibodies for protection. Since neutralizing activity is correlated with vaccine protection against symptomatic infection and vaccine efficacy, these data support the current recommendations for vaccination and boosting, even for those previously infected, to maintain the highest level of antibodies and protection.

## Methods

### Patients
This research complies with all relevant ethical regulations and was approved by the Weill Cornell Medicine Institutional Review Board. The study uses leftover specimens for routine clinical care that would have been otherwise discarded and patient information was de-identified at time of collection. A waiver of informed consent was granted by the Weill Cornell Medicine Institutional Review Board. Patients that delivered at an academic medical center in New York City between April 18, 2020, and April 27, 2022, were included in this study. Patients were included if they had documented receipt of at least one dose of a SARS-CoV-2 vaccine; Pfizer-BioNTech (BNT162b2), Moderna (mRNA-1273), or Johnson&Johnson (Ad26.COV2.S), or a history of SARS-CoV-2 infection. The vaccination type, number of doses, and dates received were abstracted from the electronic medical record. Full course of vaccination was categorized as 1 dose for the Johnson&Johnson vaccine, 2 doses for the Pfizer-BioNTech vaccine, and 2 doses for the Moderna vaccine. Patients were designated as either not having received any vaccination (NoVx), having completed a full course of vaccination if they were at least 14 days post the last dose (FullVx), having started a vaccination course but do not meet criteria for FullVx (PartVx), or having received an additional booster dose after completion of a full course of vaccination (for a total of 2 doses for a Johnson&Johnson original dose, or a total of 3 doses for Pfizer-BioNTech or Moderna original dose) (BoostVx). Patients who received a booster dose were confirmed to not have any immunosuppressing condition or use of an immunosuppressing medication. History of past infection (Inf) was identified through the electronic medical record for history of positive SARS-CoV-2 clinical tests and confirmed with detection of anti-Nucleoprotein (N) antibodies that are made only in the setting of an infection and not vaccination. Date of the infection (if available) was abstracted from the electronic medical record. Age matched patients delivering during the same period with no history of SARS-CoV-2 infection or vaccination were included as controls and confirmed to be SARS-CoV-2 antibody negative. Demographic data were also abstracted from the electronic medical record.

### Antibody studies
Antibody studies were performed from time of delivery using both the discarded clinical maternal peripheral blood samples and the corresponding discarded clinical neonatal umbilical cord blood samples.

Immunoglobulin G (IgG) antibodies against SARS-CoV-2 were assayed using a fluorescence-based reporting system which allows for semi-quantitative detection of anti-SARS-CoV-2 spike antibodies using the clinical testing Pylon 3D platform (ET HealthCare, Palo Alto, CA). This platform utilizes a fluorescence-based reporting system which allows for semi-quantitative detection of anti-SARS-CoV-2 IgG with a specificity of 98.8%. Anti-Nucleocapsid (anti-N) antibodies were also measured with the Elecsys Anti-SARS-CoV-2 assay (Roche) to assess evidence of past SARS-CoV-2 infection. IgG levels were plotted as $\log_2 + 1^{16,39}$.

### Neutralization assays
Neutralization assays were performed on a representative cohort of 259 patients with history of SARS-CoV-2 infection and who had enough specimen leftover to perform such assays. This cohort was also selected based on the documentation of a specific date of COVID-19 infection diagnosis which allowed to deduce the variant most likely responsible for infection.

A total of 293 T (ATCC, CRL-11268) and HT1080 (ATCC, CCL-121) cells were cultured in DMEM supplemented with 10% fetal calf serum (Sigma) and 10 μg/ml gentamicin at 37 °C and 5% $CO_2$. Cells were periodically checked for mycoplasma and retrovirus contamination by DAPI staining and reverse transcriptase assays, respectively. Derivatives of HT1080 cells expressing ACE2 were generated by transducing cells with a retroviral vector expressing human ACE2 and blasticidine. Following selection, single-cell clones were derived by limiting dilution from the bulk populations and are designated HT1080/ACE2cl.14[40]. The env-inactivated HIV-1 reporter construct pHIV-1NL4-3 ΔEnv-NanoLuc was generated from pNL4-3[41]. The human codon-optimized NanoLuc luciferase reporter gene (Nluc; Promega) was inserted in place of nucleotides 1–100 of the nef gene. Thereafter, a 940-bp deletion and frameshift was introduced into env, immediately 39 to the vpu stop codon. pSARS-CoV-2 protein expression plasmids containing a C-terminally truncated SARS-CoV-2 S protein s(pSARS-CoV-2Δ19) were generated. They were derived by insertion of a synthetic human codon-optimized cDNA encoding SARS-CoV- 2 Wuhan-hu-1 or BA.1 or BA.5 spikes lacking the C-terminal 19 codons into pCR3.1. and including a substitution that inactivates the furin cleavage site (R683G). The amino acid substitutions in variant spikes relative to the parental wuhan-hu-1 are: Omicron BA.1: A67V, Δ69-70, T95I, G142D, Δ143-145, Δ211, L212I, ins214EPE, G339D, S371L, S373P, S375F, K417N, N440K, G446S, S477N, T478K, E484A, Q493K, G496S, Q498R, N501Y, Y505H, T547K, D614G, H655Y, H679K, P681H, N764K, D796Y, N856K, Q954H, N969H, N969K, L981F. Omicron BA.5: T19I, L24S, del25-27, del69-70, G142D, V213G, G339D, S371F, S373P, S375F, T376A, D405N, R408S, K417N, N440K, L452R, S477N, T478K, E484A, F486V, Q498R, N501Y, Y505H, D614G, H655Y, H679K, P681H, N764K, D796Y, Q954H, N969K[40,42,43].

Neutralization of HIV-1-based viruses expressing the NanoLuciferase gene and pseudotyped with the selected spike proteins was performed as previously described[40]. Briefly, serial dilutions of plasma samples from pregnant mothers and neonates were incubated with a fixed dose of each virus for 1 h at 37 °C. The mix was then added to HT1080-ACE2 cells, seeded at $10^4$ cells/well the day prior to infection. 48 hours post-infection cells luciferase activity was determined using the NanoGlo Luciferase Assay System (Promega) with the Glomax Navigator (Promega). The relative luminescence units were normalized to those derived from cells infected with each pseudotyped virus

in the absence of plasma. The half-maximal neutralization titers for each plasma (NT50) were determined using four-parameter nonlinear regression (least squares regression without weighing; constraints top=1 and bottom=0, using GraphPad Prism software). The average of 2-3 independent experiments was used. Neutralizing activity levels were plotted as $\log_{10} + 1$.

## Statistical analyses

Maternal antibodies and neonatal cord blood antibody levels were plotted as a function of weeks elapsed from the last dose of vaccination to birth admission. Antibody levels between different vaccination cohorts were analyzed using the Wilcoxon-Rank Sum test. Antibody levels between the no history of infection and positive history of infection groups were analyzed using the Wilcoxon-Rank Sum test. Relationship between maternal antibodies and the number of weeks elapsed from the last dose of vaccination was examined across patient groups (separated based on vaccination status and history of infection status) using linear regression and mixed ANOVA. Comparison of neutralizing activity between differentially vaccinated cohorts was done using Wilcoxon Rank Sum test. Correlation of total anti-S IgG levels to neutralization activity was determined using spearman correlation analysis and linear regression. Correlation of maternal and neonatal IgG levels and neonatal neutralization activity were determined using spearman correlation analysis and linear regression. The analysis of anti-S IgG levels includes data from 1753 patients previously published[15,16,39]. Statistical analyses were performed using R 4.1.0 RStudio 1.1.463.

## Reporting summary

Further information on research design is available in the Nature Portfolio Reporting Summary linked to this article.

## Data availability

All data generated in this study are provided in the Source Data file. Source data are provided with this paper.

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

## Acknowledgements

The work was supported by the grants Weill Cornell Covid-19 research grant- 87000783 (to Y.J.Y., L.E.R., M.P., M.J.); The Bender Foundation, Inc (to Y.J.Y., L.E.R., M.P.); CTSC Pilot Award- 5310002720 (Y.J.Y., L.E.R., M.P.); Weill Cornell Maternal Infant COVID-19 vaccination research grant- 61500602 (Y.J.Y., L.E.R., M.P.); P01AI165075 (T.H., P.B.); HHMI (P.B.).

## Author contributions

Conceptualization: E.A.M., M.P., P.B., L.E.R., T.H., Y.J.Y.; Methodology: E.A.M., C.G.C., A.C.S., D.J.P., M.J., T.K., M.C., E.B., P.B., T.H., Y.J.Y.; Investigation: E.A.M., C.G.C., A.C.S., D.J.P., I.M., T.K., S.S., M.C., E.B., A.H., E.T., D.E., T.H., Y.J.Y.; Data analysis: E.A.M., T.H., Y.J.Y.; Visualization: E.A.M., T.H., Y.J.Y.; Supervision: E.A.M., P.B., L.E.R., T.H., Y.J.Y.; Writing original draft: E.A.M., P.B., T.H., Y.J.Y.; Writing-review and editing: E.A.M., M.P., M.J., T.K., P.B., L.E.R., T.H., Y.J.Y.

## Competing interests

Dr. Riley serves as a writer for UptoDate, an editorial board member for Contemporary Ob/Gyn and the New England Journal of Medicine and advisor to MAVEN. Drs. Riley, and Prabhu serve as faculty CME educators on CMV with Medscape. Dr. Bieniasz has provided consulting services to Pfizer in the area of SARS-CoV-2 vaccines. The remaining authors declare no competing interests.
