## [Peer Review File · Nature Communications]

SARS-CoV-2 vaccination, booster, and infection in pregnant population enhances passive immunity in neonatesReviewers' Comments:

Reviewer #1:

Remarks to the Author:

The manuscript titled "SARS-CoV-2 vaccination, booster, and infection enhances passive immunity" describes a retrospective hospital-record based study in NYC on the immune response following vaccination against SARS-CoV-2 in a large cohort (N= 4600) of pregnant women and their infants.

The anti-spike neutralizing Ab immune response in pregnant women in relation to prior history of infection as well as with regard to neutralization of Omicron variants was similar to what has previously been shown for non-pregnant adults. The data confirm the benefit of vaccinating pregnant women against SARS-CoV- based on increased neutralizing Ab levels. Of note, the clinical benefit of vaccination has been shown previously in large studies in pregnant women in several countries such as Israel, the UK , and the US.

The title of the study should reflect the study population included in the study, not the general target population.

The study does not differentiate/stratify study results by type of vaccine (platform) received. This analysis would be of interest, particularly because previous studies have shown differential vaccine-induced immune responses in pregnant women (Atyeo C, 2022).

The retrospective design of the study introduces inaccuracies regarding timing of infection. Gestational age at vaccination is another important predictive factor and possible confounder for the level of passively transferred Ab to the fetus that was not analyzed. This should be included in the discussion section.

The study provides novel data on the effect of prior maternal infections and vaccination history on the level of neutralizing Ab in their infants. These data provide important insights to inform decisions supporting the inclusion of pregnant women in booster vaccine recommendations.

Reviewer #2:

Remarks to the Author:

This study analyzed both the binding and neutralizing antibody responses in a large cohort of pregnant patients with a heterogeneous history of SARS-CoV-2 vaccination and infection. Cord blood from part of the participants was also analyzed to investigate the passive immunity from vaccination and infection. The manuscript is well written and the data are clearly presented.

- This study compared patients with and without an infection history. However, the infected groups contain participants who got infected at different timepoints, ie, infection before dose 1, after dose 1, after dose 2 or after booster. If a patient got infected after dose 2, the PartVx and FullVx results should be in the same group as those without an infection history. Putting all the infected patients in one group regardless of the infection time will cause bias and decrease the actual effect of infection on antibody levels. Although the precise dates of infection were not available, the infection period relative to the vaccine doses should be more accessible.

- Line 36, "In a subset of patients", please provide the exact number here. Line 40, please also give the number of patients.

- Figure 1A and 1D are basically the same figures, just the PartVx and PartVx /Inf are not included in 1D. Could you take out 1D and put the regression lines in 1A?

- Line 215, "All 8 patients with the highest neutralizing antibody levels". Are the antibodies against B.1?
- Line 215-220, the patients and the results couldn't be identified in Figure 2A as referred to. Marking the dots in Figure 2A or providing a supplemental figure will be helpful.
- Line 223, "... 96% (117/122) were born to unvaccinated or partially vaccinated patients (Figure 3A)". Were unvaccinated patients included in Figure 3A? What color and dots represented them?
- Figure 3, it would be better to label the left panel as A, B and C individually, then D, E and F for the right panel
- Could you provide some description of the bottom figure in Figure 3A? It was not mentioned in the text.
- It included both IgG and neutralizing antibody results in Figure 3, so should add IgG to the figure title.
- Line 269-273, please provide citations.
- Table 1, it's better to divide the participants into groups to make it more clear and related to the figures, for example, no vaccination and no infection, vaccination without infection, and vaccination with infection.

Reviewer #1 (Remarks to the Author):

The manuscript titled "SARS-CoV-2 vaccination, booster, and infection enhances passive immunity" describes a retrospective hospital-record based study in NYC on the immune response following vaccination against SARS-CoV-2 in a large cohort (N= 4600) of pregnant women and their infants.

The anti-spike neutralizing Ab immune response in pregnant women in relation to prior history of infection as well as with regard to neutralization of Omicron variants was similar to what has previously been shown for non-pregnant adults. The data confirm the benefit of vaccinating pregnant women against SARS-CoV- based on increased neutralizing Ab levels. Of note, the clinical benefit of vaccination has been shown previously in large studies in pregnant women in several countries such as Israel, the UK, and the US.

The title of the study should reflect the study population included in the study, not the general target population.

We thank you for your recommendation and apologize for the oversight. We have adjusted the title to: 'SARS-CoV-2 vaccination, booster, and infection in pregnant population enhances passive immunity.'

The study does not differentiate/stratify study results by type of vaccine (platform) received. This analysis would be of interest, particularly because previous studies have shown differential vaccine-induced immune responses in pregnant women (Atyeo C, 2022).

We thank the reviewer for this suggestion and we greatly valued reading and following the Atyeo et al. paper which has driven our interests in further sub analysis. We have included the requested analyses of both the patients and the neonates and included the findings into the texts for both IgG levels and neutralization activity. These data are also newly included as Figures 2, 4, and 6. In Addition, the findings were added into the discussion.

"Analyses of differences in IgG levels elicited between vaccination types showed that there was only a significant difference between receipt of mRNA1273 (Moderna) vaccine or the BNT162b2 (Pfizer-BioNTech) vaccine in the FullVx ($p=1.4e-11$) and BoostVx ($p=6.4e-8$) populations without a history of infection, but not in any of the infected patient populations (all $p>0.05$) (Fig. 2a-b)."

"Further sub-analyses of our cohort showed there were no differences detected in neutralizing activity when comparing between Moderna and Pfizer-BioNTech vaccination recipients (all $p>0.05$) (Fig. 4 a-c)."

"Additional sub-analyses showed that there were also no differences in umbilical cord blood neutralizing activity between vaccination types received (Fig. 6a-c)."

"For patients with a history of infection, we did not see a difference in neutralizing activity against B.1, BA.1, and BA.5 between vaccination type received in either the maternal peripheral blood or neonatal cord blood."

Fig. 2

Figure 2: Maternal anti-S IgG activity across SARS-CoV-2 vaccine platforms in patients with and without positive history of infection

a. Comparing maternal anti-S IgG response between vaccine platforms mRNA1273 (Moderna, purple) and BNT162b2 (Pfizer-BioNTech, orange) in patients without a history of infection stratified by vaccination status at time of delivery; Partially vaccinated without a history of infection (PartVx), Fully vaccinated without a history of infection (FullVx), Boosted without a history of infection (BoostVx). Significance was determined by Wilcoxon ranked Sum Test.

b. Comparing maternal anti-S IgG response between vaccine platforms mRNA1273 (Moderna, purple) and BNT162b2 (Pfizer-BioNTech, orange) in patients with a positive history of infection stratified by vaccination status at time of delivery; Partially vaccinated with a positive history of infection (PartVx/Inf), Fully vaccinated with a positive history of infection (FullVx/Inf), and Boosted with a positive history of infection (BoostVx/Inf). Significance was determined by Wilcoxon ranked Sum Test.

Fig. 4

Figure 4: Comparing neutralizing activity against variants across vaccine platforms in vaccinated patients

a-c. Neutralizing activity against (a) B.1, (b) BA.1, and (c) BA.5 variants in patients with a positive history of SARS-CoV-2 infection compared across vaccine platforms mRNA1273 (Moderna, purple) and BNT162b2 (Pfizer-BioNTech, orange). Patients are stratified by vaccination status at time of delivery; Partially vaccinated with a positive history of infection (PartVx/Inf), Fully vaccinated with a positive history of infection (FullVx/Inf) and Boosted with a positive history of infection (BoostVx/Inf). Significance was determined by Wilcoxon ranked Sum Test.

Fig. 6

Figure 6: Comparing neutralizing activity against variants across vaccine platforms in neonates of vaccinated patients

a-c. Neutralizing activity against (a) B.1, (b) BA.1, and (c) BA.5 variants in umbilical cord blood from patients with a positive history of SARS-CoV-2 infection compared across vaccine platforms; mRNA1273 (Moderna, purple) and BNT162b2 (Pfizer-BioNTech, orange). Samples are stratified by patient vaccination status at time of delivery; Partially vaccinated with a positive history of infection (PartVx/Inf), Fully vaccinated with a positive history of infection (FullVx/Inf) and Boosted with a positive history of infection (BoostVx/Inf). Significance was determined by Wilcoxon ranked Sum Test.

The retrospective design of the study introduces inaccuracies regarding timing of infection. Gestational age at vaccination is another important predictive factor and possible confounder for the level of passively transferred Ab to the fetus that was not analyzed. This should be included in the discussion section.

We agree with the points that the reviewer brought up and thank you for the suggestion. Our patient population is largely consistent with carrying to term with a median (IQR) GA at birth of 39.3 (1.5), therefore the weeks elapsed from last dose to delivery may serve as a proxy for gestational age at vaccination. In addition, given the long time frame for when patients receive their primary course and the delay until booster, the patients may have received certain doses before pregnancy and others during pregnancy. We have inserted all of these caveats into our discussion per reviewer recommendations.

“Given the recommended spacing of months between the primary vaccination course and the booster dose, it possible that the boosted cohort started vaccination earlier than the non-boosted cohort; however, our analysis has been performed with respect to the timing of the last vaccination dose received prior to birth and we have ensured that there is a good spread of patients with comparable demographics in all cohorts tested to the best of our ability. Our study did not include analysis on the gestational age at any singular vaccination receipt which is a predictive factor that could be playing a role in the level of passive transfer of antibodies; however, our patient population has a median (IQR) gestational age at birth of 39.3 (1.5) weeks and our analysis on weeks elapse from last vaccination to birth may serve as a proxy for understanding this possible confounding variable.”

The study provides novel data on the effect of prior maternal infections and vaccination history on the level of neutralizing Ab in their infants. These data provide important insights to inform decisions supporting the inclusion of pregnant women in booster vaccine recommendations.

We thank the reviewer for this kind comment acknowledging the importance of this work.

Reviewer #2 (Remarks to the Author):

This study analyzed both the binding and neutralizing antibody responses in a large cohort of pregnant patients with a heterogeneous history of SARS-CoV-2 vaccination and infection. Cord blood from part of the participants was also analyzed to investigate the passive immunity from vaccination and infection. The manuscript is well written and the data are clearly presented.

This study compared patients with and without an infection history. However, the infected groups contain participants who got infected at different timepoints, ie, infection before dose 1, after dose 1, after dose 2 or after booster. If a patient got infected after dose 2, the PartVx and FullVx results should be in the same group as those without an infection history. Putting all the infected patients in one group regardless of the infection time will cause bias and decrease the actual effect of infection on antibody levels. Although the precise dates of infection were not available, the infection period relative to the vaccine doses should be more accessible.

We thank the reviewer for this comment. We have earnestly tried to sub-stratify the population using this exact strategy since the very beginning of our studies and data gathering, as knowing the dates of vaccination would have been valuable to also understanding the potential variants of infection and allow us to gain many more additional layers of insights from such a unique large cohort. However, with all of our efforts--including manual curation of all patient records spanning back to early 2020 at the time of pandemic onset--it became clear that the precise dates were not available for many patients, and even the relative infection period were often times either not accessible or had inaccuracies negating our ability to feel confident enough to sub stratify the infected cohort into when they were infected with respect to when they were vaccinated. That is why the cohort that we did test with documented clear dates of infection is a small cohort focused on the ones where we were confident we could trust the dates listed based on the clinical scenario/history documented.

We agree that grouping all patients with history of infection together mingles patients that got infected before vaccination vs. patients that got infected post vaccination. Our goal was not to make a comment on the effects of infection on vaccination in a sequential manner, but to elucidate the fact that there are still benefits to vaccination or boosting even if there is a history of infection irrespective of when you may have been infected, in a continued effort to quell vaccination and boosting hesitancy in the patient population. We also found that this echoes some of what we see in our patient population which is that many may not even have knowledge of when they were infected, while others may even have gotten Covid-19 multiple times.

We have included a discussion on this potential caveat to the manuscript.

“Limitations in our study include the dependence on self-reported history of infection and a fraction of the infected cohort lacked precise dates of infection or viral sequencing, and therefore we cannot deduce the variant responsible for the infection. The lack of precise timing of infection may have hindered our ability to elucidate significant association between time elapsed since infection and antibody levels as well as the potential variants of infection. In addition, the lack of timing did not allow us to sub stratify patients based on when the vaccinations and infections took place relative to one another; thus, grouping of all infected patients cannot address the direct effect of infection on antibody levels from vaccination or the direct effect of vaccination on antibody levels from infection.”

Line 36, “In a subset of patients”, please provide the exact number here. Line 40, please also give the number of patients.

We thank the reviewer for their suggestion and have amended the text to include the number of patients studied.

Figure 1A and 1D are basically the same figures, just the PartVx and PartVx /Inf are not included in 1D. Could you take out 1D and put the regression lines in 1A?

We appreciate this great catch/recommendation and have edited figure 1, legend, and text to reflect these changes.

Line 215, “All 8 patients with the highest neutralizing antibody levels”. Are the antibodies against B.1?

We thank the reviewer for this thoughtful question. Yes, the antibodies are against B.1 and the text has been edited to clarify the variant of interest for future readers.

“All 8 patients with the highest B.1 neutralizing antibody levels...”

Line 215-220, the patients and the results couldn't be identified in Figure 2A as referred to. Marking the dots in Figure 2A or providing a supplemental figure will be helpful.

We greatly appreciate the reviewer's suggestion and have provided a supplemental figure to highlight the data points as recommended (see supplementary figure 1). Each group is represented by a distinct shape on the supplementary figure; the population of patients with high B.1 neutralizing antibody levels (n = 8), patients with low B.1 levels (n = 35), and patients with high omicron levels (n = 50). Thank you very much for the comment and opportunity to provide more clarity with the text.

Line 223, “... 96% (117/122) were born to unvaccinated or partially vaccinated patients (Figure 3A)”. Were unvaccinated patients included in Figure 3A? What color and dots represented them?

We sincerely thank the reviewer for catching this carry-forward error from when we were considering this analysis with the unvaccinated/infected patients. You are correct that there were no unvaccinated patients included in Figure 3A (now Fig 5a, 5b, and 5c). The text now reads as, “56 umbilical cord samples had IgG levels below the positive cutoff, and of those 56 samples, 91% (51/56) were born to partially vaccinated patients (Fig. 5b).” The partially vaccinated patients are designated in green. We have confirmed that there are no errors in the data included or represented in the figures.

Figure 3, it would be better to label the left panel as A, B and C individually, then D, E and F for the right panel

We appreciate the recommendation to reformat and have changed Figure 3 (now Figure 5), legend, and text to based on the suggestions.

Could you provide some description of the bottom figure in Figure 3A? It was not mentioned in the text.
The transfer ratio

We thank the reviewer for their comment and apologize for the oversight. We have provided an additional sentence describing the bottom of figure 3A (now figure 5C based on changes from previous suggested edit), “The placental transfer ratio was above 0.5 for 93% (1785/2429) of all dyads tested (Fig. 5c).”

It included both IgG and neutralizing antibody results in Figure 3, so should add IgG to the figure title.

We thank the reviewer for catching this omission and have added IgG to the description of Figure 3 (now Figure 5).

Line 269-273, please provide citations.

We appreciate the reviewer's comment and we have included the appropriate citations as requested.

Table 1, it's better to divide the participants into groups to make it more clear and related to the figures, for example, no vaccination and no infection, vaccination without infection, and vaccination with infection.

We thank the reviewer for this suggestion. We have provided the requested additional Table with participant subgroups more clearly represented. After discussion, we left both tables in as it allowed for both a global view of the patient population as a full cohort as well as the ability to understand the groups included in each figure. The two tables are included as Table 1 and Supplementary Table 1 with references in the text. A breakdown of the population where neutralization studies were performed was also included as Supplementary Table 2.

Reviewers' Comments:

Reviewer #1:

Remarks to the Author:

All comments have been adequately addressed.

Reviewer #2:

Remarks to the Author:

The authors have addressed all my questions. I have one more question out of interest, what is the reason for changing the cutoff of the transfer ratio from 1.0 to 0.5 (figure 5C, previous figure 3A)? It's easier to understand that if the ratio is greater than 1, the antibody levels in the cord blood are higher than that in the maternal blood, which indicates efficient transfer. The cutoff of 1.0 is also more commonly used for transplacental transfer studies. Lowering the cutoff increases the positive rate (more participants are above the cutoff), but are there other interpretations for reporting the ratio at 0.5?

Other than this, the manuscript is ready for publication.

Reviewer #1 (Remarks to the Author):

All comments have been adequately addressed.

We thank the reviewer for all comments and appreciate their time spent assessing and improving our manuscript.

Reviewer #2 (Remarks to the Author):

The authors have addressed all my questions. I have one more question out of interest, what is the reason for changing the cutoff of the transfer ratio from 1.0 to 0.5 (figure 5C, previous figure 3A)? It's easier to understand that if the ratio is greater than 1, the antibody levels in the cord blood are higher than that in the maternal blood, which indicates efficient transfer. The cutoff of 1.0 is also more commonly used for transplacental transfer studies. Lowering the cutoff increases the positive rate (more participants are above the cutoff), but are there other interpretations for reporting the ratio at 0.5?

Other than this, the manuscript is ready for publication.

We thank the reviewer for this discussion. We have been fortunate to see immediate feedback from patients and the interpretation of the general public to our previous publications. The possible protection offered to the baby from maternal vaccination is a strong motivator for vaccination in pregnant patients. Our goal was to express the transfer ratio as a pure descriptor of the data rather than an inference of clinical impact (i.e. protection at a certain transfer ratio or lack thereof below); however, we were careful with also balancing the fact that we didn't want a low % listed to be interpreted incorrectly or as something "less efficacious." We were especially careful with this given the points made by the reviewer. In an effort to maintain the "description" aspect of the data and of this statement, we have listed the % transfer ratio at both 0.5 and 1.0 and have added the line.